# Pancreatic Cancer in Chronic Pancreatitis: Pathogenesis and Diagnostic Approach

**DOI:** 10.3390/cancers15030761

**Published:** 2023-01-26

**Authors:** Guillaume Le Cosquer, Charlotte Maulat, Barbara Bournet, Pierre Cordelier, Etienne Buscail, Louis Buscail

**Affiliations:** 1Department of Gastroenterology and Pancreatology, CHU Toulouse-Rangueil, University Hospital Centre, Toulouse University, UPS, 31059 Toulouse, France; 2Institut de Recherche en Santé Digestive, Toulouse University, INSERM U1022, INRAe, ENVT, 31300 Toulouse, France; 3Department of Digestive Surgery, CHU Toulouse-Rangueil, University Hospital Centre, Toulouse University, UPS, 31059 Toulouse, France; 4Toulouse Cancer Research Center, INSERM U1037, Toulouse University, 31100 Toulouse, France; 5Centre for Clinical Investigation in Biotherapy, CHU Toulouse-Rangueil, University Hospital Centre, INSERM U1436, 31059 Toulouse, France

**Keywords:** chronic pancreatitis, pancreatic ductal adenocarcinoma, screening, alcohol consumption, tobacco consumption, hereditary pancreatitis

## Abstract

**Simple Summary:**

Chronic alcoholic pancreatitis displays a cumulative risk of pancreatic cancer estimated at 4% after 15 to 20 years, this risk being higher for hereditary pancreatitis with 19% and 12% in the case of *PRSS1* and *SPINK1* mutations, respectively, and at an age of 60 years. Oncogene and tumor suppressor gene mutations associated with chronic inflammation are key promoters of this complication, tobacco being an additional co-factor. This event underlines two practical problems from a clinical point of view: diagnosis is difficult due clinical symptoms and radiological features shared by the two diseases; and screening of cancer in chronic pancreatitis patients. Endoscopic ultrasound-guided fine-needle biopsy can be contributive with the help of molecular biology by next generation sequencing, including for *KRAS, TP53, CDKN2A,* and *DPC4* mutation assays. A short-term follow-up of patients is necessary in cases with clinical and/or radiological suspicion of cancer. Pancreatic surgery is sometimes necessary if there is any doubt.

**Abstract:**

Chronic pancreatitis is one of the main risk factors for pancreatic cancer, but it is a rare event. Inflammation and oncogenes work hand in hand as key promoters of this disease. Tobacco is another co-factor. During alcoholic chronic pancreatitis, the cumulative risk of cancer is estimated at 4% after 15 to 20 years. This cumulative risk is higher in hereditary pancreatitis: 19 and 12% in the case of *PRSS1* and *SPINK1* mutations, respectively, at an age of 60 years. The diagnosis is difficult due to: (i) clinical symptoms of cancer shared with those of chronic pancreatitis; (ii) the parenchymal and ductal remodeling of chronic pancreatitis rendering imaging analysis difficult; and (iii) differential diagnoses, such as pseudo-tumorous chronic pancreatitis and paraduodenal pancreatitis. Nevertheless, the occurrence of cancer during chronic pancreatitis must be suspected in the case of back pain, weight loss, unbalanced diabetes, and jaundice, despite alcohol withdrawal. Imaging must be systematically reviewed. Endoscopic ultrasound-guided fine-needle biopsy can contribute by targeting suspicious tissue areas with the help of molecular biology (search for *KRAS, TP53, CDKN2A, DPC4* mutations). Short-term follow-up of patients is necessary at the clinical and paraclinical levels to try to diagnose cancer at a surgically curable stage. Pancreatic surgery is sometimes necessary if there is any doubt.

## 1. Introduction

Chronic calcifying pancreatitis (CP) is a chronic inflammation of the pancreas associated with the development of fibrosis of the parenchyma. This inflammation leads to progressive and irreversible focal, segmental, or diffuse lesions within exocrine and endocrine pancreatic tissue. At the end, extensive fibrosis occurs with destruction of the exocrine tissue, pancreatic ducts, and islets of Langerhans. This fibrosis is associated with protein plugs which are responsible for the calcifications visible on imaging and which may obstruct the pancreatic ducts. Progressive damage to the pancreas generates clinical manifestations with the occurrence of acute pancreatitis, pain, exocrine pancreatic insufficiency, and diabetes [1,2,3,4].

In terms of etiology, alcohol is the main agent, with tobacco consumption as an important co-factor. There are also genetic (including hereditary chronic pancreatitis due to mutation of the *PRSS1* gene but also other forms due to mutations of other genes such as *CFTR, SPINK1, TRPV6,* and *CTRC*) and idiopathic forms. Two particular forms should be set apart: autoimmune pancreatitis and chronic obstructive pancreatitis. In the case of autoimmune pancreatitis, pathophysiology, lesions, natural history, and treatment are very different from classical CP. Obstructive CP develops upstream of an obstruction/stenosis in the Wirsung duct, due to a malignant or benign ampullary or pancreatic tumor, including intraductal papillary mucinous neoplasms (IPMN).

CP evolves in three successive phases in terms of clinical manifestations and complications [2,3,4]. The first five years are rich in clinical manifestations with chronic pain of pancreatic origin, followed by attacks of acute pancreatitis and its own complications (serous effusions, stenosis of the main bile duct). At this stage, exocrine and/or endocrine pancreatic insufficiency only concerns one third of patients. Between 5 and 10 years of evolution, patients still present painful phenomena, but these are much less frequent, as are attacks of acute pancreatitis. On the other hand, pseudocysts, serous effusions, and stenosis of the main bile duct are still present. Finally, after 10 years of CP evolution, the painful phenomena disappear as the pancreas becomes totally fibrous, without inflammation or possible enzymatic activation. At this stage, exocrine pancreatic insufficiency and diabetes are predominantly present. At this time, the risk of pancreatic ductal adenocarcinoma (PDAC) increases. It is obvious that the times and periods proposed in this natural history schema will depend on the degree of alcohol consumption but also on tobacco consumption [5]. Indeed, the persistence of either form of intoxication accelerates parenchymal and ductal damage. In particular, even in the presence of prolonged alcohol withdrawal, continued smoking alone can cause further progression of ductal and parenchymal lesions [6,7].

In addition to this evolution in the patient’s clinical symptomatology, there is a progressive change in the imaging appearance. In fact, at the beginning, calcifications are absent and classical imaging (CT scan or MRI cholangio-pancreatography) poorly displays pancreatic lesions. Endoscopic ultrasound (EUS) plays an important role in the diagnosis of these early, non-calcified forms of CP. Subsequently, we see more clearly either in MRI or CT scans an increase in ductal irregularity with the appearance of stenosis(s) and dilatation(s) and an increasingly heterogeneous parenchyma.

CP is a risk factor for pancreatic ductal adenocarcinoma (PDAC). This is a difficult and relevant problem from a pathophysiological, epidemiological, and clinical point of view. Clinical approach is peculiarly delicate and both positive and differential diagnosis of PDAC developed on CP is a real challenge due to the rather low specificity of symptoms, imaging signs, and biological markers. Beyond diagnosis, there is also the problem of screening. All of these points will be addressed in this review in an attempt to draw practical conclusions and provide advice. In other terms, why can cancer develop during chronic pancreatitis and how is it diagnosed?

## 2. How Does Pancreatic Adenocarcinoma Develop from Chronic Pancreatitis?

### 2.1. Genetics of Pancreatic Carcinogenesis

A large number of genes are altered during PDAC implicated in various molecular functions such as MYC activation, TGFβ signaling, G1/S checkpoint, genome stability, Wnt/notch signaling, RNA splicing, homologous recombination deficiency, and KRAS signaling [8,9,10,11,12]. An activating point mutation of the *KRAS* oncogene on codon 12 (exon 2) is the initiating event in the majority of PDAC cases (70–95%) [8,9,10,11,12]. Point mutations can also occur, although less frequently, on codons 11, 13, 61, or 146 of the *KRAS* oncogene [13,14]. The point mutation of *KRAS* impairs the intrinsic GTPase activity of RAS protein and prevents GTPase-activating proteins from promoting the conversion of GTP (active) to GDP (inactive). The RAS protein is thus permanently bound to GTP and constitutively activates downstream effector proteins and signaling pathways (such as EGFR signaling) as well as nuclear factors, thereby leading to stimulation of cell differentiation, proliferation, migration, transformation, adhesion, and survival [10,11,14]. In addition, whereas oncogenic *KRAS* is activated, *INK4a*, *TP53*, and *DPC4* tumor suppressor genes are epigenetically or genetically inactivated in the majority of PDAC [8,11,15]. The *KRAS* mutation occurs early in pancreatic carcinogenesis as attested by its presence in common pre-neoplastic and precursor lesions, such as pancreatic intraepithelial neoplasia (PanINs) and IPMNs [8,10,11,13]. PanINs are developed from pancreatic duct cells with a hyperplastic shape and a preneoplastic potential. They are classified as PanIN-1 to PanIN-3 depending on the degree of cytological atypia, with PanIN-3 being the last stage before PDAC. If *KRAS* mutation appears early in pancreatic carcinogenesis (PanIN-1), PanINs display progressively genetic alterations, such as loss of heterozygosity and function of *P16* (*INK4a–ARF*), *TP53*, and *DPC4–SMAD4* [10,11]. These genetic events appear according to the grade of dysplasia confirming the multi-step status of pancreatic carcinogenesis at the genetic level and the critical role of PanINs in PDAC initiation.

### 2.2. Pancreatic Carcinogenesis during Chronic Pancreatitis: Role of Acinar Ductal Metaplasia

Chronic pancreatitis and PDAC were historically considered as unrelated diseases as they were thought to arise from two different cells in the pancreas, namely acinar and ductal cells. Based on experiments using mouse models, a common origin in acinar cells has been proposed. The model of PDAC was thought to develop from the duct cells via PanINs lesions [12]. However, this progression model of PanINs-PDAC has been questioned regarding the molecular status of the transgenic model of PDAC in mice. Indeed, mouse lineage lacked evidence of oncogenic *KRAS* activation in pancreatic ductal cells while PanINs and PDAC were commonly observed in these models [16]. However, the mouse model introduces the concept of acinar to ductal metaplasia that will precede the generation of small ducts. Indeed, acinar to ductal metaplasia has been well documented in experimental rodent models of pancreatitis [16]. On the other hand, treatment with the cholecystokinin agonist, cerulein, induces local oxidative stress, inflammation, edema, and loss of acinar parenchyma. The latter is transiently replaced by duct-like epithelium, thus approximating the human pancreatitis model [15,16,17]. Secondly, evidence exists that mouse models support the idea that activation of oncogenic *KRAS* specifically in acinar cells during embryonic development induces the formation of PanINs and the development of invasive ductal carcinoma [8,11,16,17,18,19,20]. All these important and converging observations led to the conclusion that pancreatic cancer can originate from acinar cells via acinar metaplasia leading to ductal metaplasia. Subsequently, a combination of genetic alterations (activation of the *KRAS* oncogene and loss of expression of tumor suppressor genes) associated with extrinsic factors induce tissue damage, such as oxidative stress and inflammatory damage as observed during pancreatitis [15,21,22].

Several closely related mechanisms can contribute to the transformation of acinar cell metaplasia into cancer: oxidative stress, activation of inflammatory pathways (such as Cyclooxygenase-2 (Cox2)), and activation of NF-κB and STAT3 pathways. Oxidative stress and the generation of reactive oxygen species and reactive nitrogen species play a key role in the pathophysiology of acute and chronic pancreatitis. They perpetuate acinar cell necrosis and fibrosis and thereby modify critical substrates, such as nucleic acids, lipids, and proteins, which results in DNA fragmentation, membrane disintegration, and protein misfolding. The immune cells and macrophages are also activated (and in turn produce cytokines and interleukins) as well as other stromal components, such as endothelial cells and pancreatic stellate cells, that in turn produce inflammatory cytokines and chemokines (such as Interleukin 6). All these compounds, together with reactive oxygen species/reactive nitrogen species, induce epithelial cell damage and increased proliferation. The inflammatory mediators, such as Cox2, NF-κB and STAT3, also play key roles as inflammation can generate sustained secondary oxidative injury, further inducing the promotion of inflammatory infiltration and acinar cell injury [16,20] (Figure 1). Thus, activation of NF-κB and STAT3 has been shown to be essential for pancreatic cancer initiation and progression [23,24]. STAT3 participates in cancer initiation by promoting dedifferentiation of acinar cells during pancreatic inflammation. As a consequence, these acinar cells become more vulnerable to transformation initiated and driven by the *KRAS* oncogene [24,25] (Figure 1).

During acute pancreatitis, there are also other processes related to the phenomenon of pyroptosis with activation of the NLRP3 inflammasome and it’s signaling pathway. This pathway includes the activation of caspase-1, IL1-β and IL-18 [21,22].

Experimental evidence shows that adult mice that endogenously express oncogenic mutated KRAS in the acinar cell develop PanIN and PDAC lesions with higher penetrance when subjected to acute or chronic inflammation induced by cerulein. [20]. Sporadic episodes of pancreatitis (e.g., one month) are sufficient to induce PDAC in these mice. Longer inflammatory episodes (three months) or the induction of chronic pancreatitis, increase the incidence of tumors and reduce the latency of onset [18,21,22,26] (Figure 2). Even more importantly, adult mice can develop PDAC after induction of mutated KRAS in epithelial cells only following treatment with cerulein [23]. Molecular exploration has further demonstrated that oncogenic *KRAS* do prevent tissue repair following acute pancreatitis (second hit theory).

Pancreatic inflammation induced by cerulein injections increases the burden of PanINs and PDAC in transgenic mice bearing KRAS and p16INK4a or Trp53 mutations.

### 2.3. Mouse Models of PDAC

The study of *KRAS* mutations has greatly improved the understanding of the processes involved in the transformation, uncontrolled proliferation, and invasion of pancreatic cancer cells. This has been made possible by the creation of the *KRASG12D* transgenic mouse model [11,12,15,21]. These mice express the *KRAS* oncogene early in the embryonic development of the pancreas. PanINs are already observed shortly after weaning and progress in grade and number over time. These mice mimic human pathology quite closely with the presence of PanINs lesions in 100% of the animals [11,27,28,29]. Several studies based on transgenic *KRAS^G12D^* mice have demonstrated the role of pro-inflammatory molecules as well as the role of inflammation in general in pancreatic carcinogenesis. As an example, transgenic overexpression of Cox2 in the pancreas induces chronic pancreatitis and the formation of pre-invasive ductal neoplasms [16,24]. Other studies have showed that in a *KRAS* mutant context, TNF-α-induced activation of the NF-κB pathway in pre-malignant epithelial cells enhances notch signaling in an Ikk2-dependent manner [25,26]. A non-exhaustive list of genetically engineered mice demonstrating the role of inflammatory genes/pathways in the development of pancreatitis, pancreatic intraepithelial neoplasia, and pancreatic cancer is presented in Table 1. All these models demonstrate the role of inflammation and its mediators in acinar to ductal metaplasia as well in the progression towards PanINs lesions and PDAC in collaboration with genetic events, i.e., early oncogenic mutation of *KRAS* and subsequent inactivation of tumor suppressor genes.

In addition to transgenic models, an original model has been created in mice, which consists of inducing pancreatic inflammation by injecting cerulein in combination with a carcinogen, azoxymethane (both intraperitoneally). This model also induces pancreatic carcinoma lesions associated with an acino-ductal metaplasia [27,28].

Taken together, this experimental evidence may partially explain the fact that acute and chronic inflammation in both pancreatic acinar cells and their surrounding stroma can lead to PDAC formation.

## 3. Incidence of Pancreatic Cancer in Chronic Pancreatitis

Alcoholic CP is a risk factor for the development of pancreatic cancer. The relative risk compared with a control population is estimated to be 1.8 to 2 at 10 years of evolution. However, the cumulative risk of cancer is 2% after 5 years and 4% after 15 to 20 years, i.e., at a late stage of the disease. As suggested above, smoking is thought to play a role as a facilitating factor with a pro-inflammatory effect on the pancreas, aggravating the lesions of alcoholic CP and thus increasing the risk of cancer [3,34,35].

Aside from alcohol, several gene variants are involved in the development of genetic chronic pancreatitis. Hereditary pancreatitis is linked to a mutation in the *PRSS1* gene which codes for the cationic trypsinogen. Its transmission mode is autosomal dominant with a penetrance of 80% (main mutations are R122H and N29I). There are other forms of genetic CP due to the mutation of “susceptibility” genes: the *SPINK1* gene (serine protease inhibitor Kazal type 1), that codes for the cationic trypsinogen inhibitor; the *CFTR* gene (cystic fibrosis transmembrane conductance regulator) that codes for the chlorine channels of the pancreatic ductal cells and other organs; the *CTRC* gene encoding Chymotrysin C; more rarely *CASR* (calcium-sensing receptor), *CLDN2* (protein claudin 2), *CPA1* (carboxypeptidase A1), *TRPV6* (TRPV6 calcium channel regulation), and the *CEL-HYB* (carboxyl ester lipase pseudogene) allele [3].

In hereditary pancreatitis (secondary to the *PRSS1* gene mutation), the risk of cancer is much higher than in alcoholic CP. In addition, there are a large number of pre-neoplastic lesions in the pancreas, in particular PanINs, which concerns 75% of patients after 25 years, including 25% with PanIN-3 (i.e., with high-grade dysplasia and in situ carcinoma) [36]. The most likely explanation is the longer duration of evolution of this form of CP and longer exposure to inflammatory phenomena, evolving since childhood. In the study by Rebours et al., based on the French national cohort of patients with a *PRSS1* gene mutation, the cumulative risk of pancreatic cancer at the age of 50, 60, and 75 years was 10%, 19%, and 53.5%, respectively. Furthermore, the risk of PDAC is estimated to be 80 times higher than in the general population [37]. In *PRSS1* mutations, the presence of concomitant other gene mutations (*CFTR* and/or *SPINK1*) were not associated with an increased risk of cancer, neither were diabetes, smoking status, or paternal inheritance of the genetic disease. Indeed, the relative risk of cancer in smokers was 8.5 with a cumulative risk of more than 50% at an age of 70 years [37]. Data from the European Registry of Hereditary Pancreatitis and Pancreatic Cancer (418 individuals included) showed similar results with a cumulative risk of PDAC estimated at 44.0% at 70 years (relative risk at 10) [38] (Table 2).

With regard to genetic chronic pancreatitis in the context of a *SPINK1* gene mutation, the recent work of Muller et al., combining the results of a French and English cohort, observed an excess risk of pancreatic cancer of 3.3% of patients when compared with a control cohort of idiopathic CP [39]. The risk of cancer (rough rate) was 0.8% before 50 years, 12% at 60 years, 28% at 70 years, and 52% at 80 years. This represents a significant 12-fold relative risk compared to patients with idiopathic CP [39]. Regarding CP related to *CFTR* gene mutation(s), the risk of PDAC is also increased compared to control patients with idiopathic CP (*p* < 0.05), the excess risk being 1.41 compared to control patients (95% confidence interval [1.07–1.84]) [40,41]. No data are available for pancreatitis due to other, otherwise less frequent, susceptibility genes. Based on the few data available, there does not seem to be an excess risk of PDAC in the case of autoimmune pancreatitis [42,43].

A study of the Danish nationwide cohort of CP estimated the over-risk of PDAC compared to controls at 6.9 (adjusted hazard ratio with 95% confidence interval of 5.6–8.6 [44]. This risk increases with disease evolution and progression, especially after 15 to 20 years. Although, clinicians should keep in mind that most PDAC diagnoses are made during the five years following the diagnosis of CP [42,45].

Ultimately, there is an increased risk in the case of hereditary pancreatitis with the confounding and aggravating factor of chronic tobacco consumption [46]. Nevertheless, considering all risk factors for PDAC (endogenous, genetic and non-genetic, exogenous) the “relative weighting” of CP involvement in all cases of PDAC does not exceed 2–7% of all PDAC [47].

## 4. Which Clinical and Paraclinical Signs in Chronic Pancreatitis Are an Indication of Cancer?

### 4.1. Clinical Signs

The diagnosis of PDAC in CP is a challenge for the clinician given the relative rarity of this event compared to other complications of CP and the shared symptomatology of PDAC with CP itself, making a clinical diagnosis difficult. Indeed, weight loss, worsening or new-onset of diabetes, and steatorrhea are suggestive of both malignant evolution and progression of CP. Regarding diabetes mellitus, a recent European study suggests that PDAC might cause new-onset diabetes mellitus (DM) while the link between long-standing DM and PDAC is not clear [48]

However, appearance of those signs or reappearance of pain in the context of cessation of pancreatotoxic consumption (tobacco, alcohol) for many years are evocative of PDAC development [3,49]. Similarly, disease duration of more than 10 years and known PRSS1 mutation should alert the clinician to the carcinomatous evolution of CP. Occurrence of jaundice with pruritus supports the diagnosis of PDAC while jaundice in CP is typically non-pruritic. Finally, the appearance of ascites suggests peritoneal carcinomatosis or the diagnosis of a pancreatic ductal rupture in case of acute pancreatitis.

### 4.2. Standard Biology

Unfortunately, standard biology and the CA 19-9 blood test, which can be elevated even in cases of purely “inflammatory” pancreatic disease, do not contribute much to the diagnosis of cancer in CP [50,51,52]. Fahrmann et al. have proposed choosing a higher cut-off for CA 19-9 in order to decrease the rate of false positives, in combination with two protein markers LRG1 and TIMP1, to increase specificity to 99% [53]. Similarly, many combinations of serological biomarkers with CA 19-9 have been proposed: PROZ, and TNFRSF6B, sTRA, and thrombospondin-2 [54,55,56]. Recently, more complex biomarker signatures combining four to eight biomarkers with CA 19-9 have been developed to obtain greater specificity and sensitivity (e.g., 92% and 99%, respectively, for the IMMray and PanCan-d test) [57,58]. Blyuss et al. have constructed a biomarker-based risk score (called PancRISK) based on urine biomarkers for stratified screening of pancreatic cancer patients. This score combined with CA 19-9 blood levels performed very well in predicting PDAC (sensitivity 0.96, specificity 0.96). Sogawa et al. have identified C4b-binding protein α-chain (C4BPA), using proteomic quantitative analysis, as a promising serum biomarker of PDAC [59]. In their study, C4BPA serum levels were significantly higher in patients with PDAC than in healthy controls and patients with pancreatitis.

In addition to these protein markers, other opportunities may emerge in the future from progress made in fundamental science. The relevance of PDAC screening might be found in the analysis of exosomes [60,61,62], miRNA expression [47,63,64,65], methylation markers [66,67], or metabolomic [68,69] or transcriptomic studies [70].

### 4.3. Imaging

#### 4.3.1. Ultrasonography, Tomodensitometry (CT Scan), and MRI

An ultrasound (US) is difficult to interpret because even if the diagnosis of CP is easily made in generally lean patients, the viewing of a tissue mass is complicated in the case of heterogeneous parenchyma. However, indirect or suggestive signs (although sometimes present in CP) such as Wirsung duct dilatation upstream of a mass, presence of lymph nodes, venous thrombosis, dilatation of the main biliary tract, presence of ascites, or suspicious intrahepatic tissue lesions should be investigated. Contrast-enhanced ultrasound brings additional data to the diagnostic process: one can observe an enhancement of PDAC tissue during the venous phase when compared to the adjacent normal pancreas. However, the major limitation of an ultrasound lies in its operator-dependent quality and interpretation. To overcome this difficulty, artificial intelligence has been proposed as a useful tool to assist radiologists [71]. For example, the model developed by Tong et al. achieved an area under the curve (AUC) of more than 0.950 in an external validation data set [72].

The CT scan must follow a precise protocol with a triphasic examination by multidetector computed tomographic angiography, with a slice thickness of ≤0.5–1 mm, without injection, then after intravenous injection of iodinated contrast product, a late arterial phase, known as the pancreatic parenchymal phase (at 40–50 s), followed by a portal venous phase (at 65–70 s). Pancreatic adenocarcinomas, both in the primary tumor and in metastases, are typically hypodense in the arterial phase and isodense/slightly enhanced in the portal phase. Multiplanar reconstructions should be performed in addition to the acquisition for vascular study. Some forms are difficult to interpret, such as more or less diffuse infiltrating forms. The aim is to highlight signs of tumor involvement, such as Wirsung duct dilation and atrophy of the pancreatic parenchyma upstream of the suspect area, venous and particularly arterial invasion (rarer in CP), metastatic lymph nodes, and peritoneal carcinosis. In addition, the infiltration of vessels and adjacent anatomical elements can take on a suspicious character. Nevertheless, these signs must be analyzed outside of acute pancreatitis attacks which can give false positive signs of tumor-like tissue infiltration. Figure 3 illustrates this point in a CT scan and an endoscopic ultrasound (EUS). The difficulty lies in the fact that in CP, the parenchyma is heterogeneous and calcified. In the case of pre-existing atrophy, the diagnosis of a developing mass will be easier because it will be possible to individualize a tissue mass or a localized infiltration. The diagnosis will be more difficult in cases of CP with multiple inflammatory, ductal, and cystic changes with multiple calcifications and a dilated Wirsung duct. Certain indirect signs may point to the diagnosis of cancer, such as peripancreatic tissue infiltration (but outside of any acute pancreatitis attack), infiltration of the vessels (particularly arterial) and the celiac region, and tissue lesions surrounded by calcifications. Similarly, in a US, artificial intelligence may significantly improve PDAC detection in the future. In the study by Liu et al., artificial intelligence outperformed analysis by radiologist in terms of sensitivity for detection of PDAC (0.983 vs 0.929; *p* = 0.014) [73]. Others have reported that perfusion CT performs better than standard CT to differentiate PDAC from mass-forming CP [74,75].

An MRI examination may have added value for the diagnosis of isodense pancreatic cancer lesions, not or poorly visible in a CT (better contrasted in MRI), and liver lesions too small to be characterized or of undetermined nature (especially in diffusion MRI) [76]. Nevertheless, the inflammatory “atmosphere” of the pancreatic parenchyma does not facilitate interpretation. Wirsung-MRI sequences should be systematically performed with analysis of the morphology of the Wirsung duct: differentiating a calcified obstruction from a stenosis that may be tumoral, differential diagnosis with chronic pancreatitis upstream of a genuine cancer [1,2,3]. The use of a secretin stimulation of pancreatic duct flow has been proposed to facilitate the analysis of duct stenosis and differentiate malignant from benign stenosis [77]. Radiomics models have been proposed to improve the radiologist’s ability to differentiate PDAC and mass-forming CP lesions [78].

Radiological diagnosis of PDAC in the context of CP is a difficult approach given the parenchymal and ductal changes seen during PC itself which in practice make it difficult to make a “morphological” diagnosis of cancer developed in PC [79]. The “morphological” aspect can also be confused with the pseudo-tumorous form of CP, particularly on the head of the pancreas in case of autoimmune pancreatitis, but also with other differential diagnoses such as para-duodenal pancreatitis and chronic obstructive pancreatitis developing upstream of Wirsung duct stenosis (primary PDAC, main or mixed IPMN otherwise often malignant) [80,81,82,83,84]. Moreover, as regards PDAC itself, Overbeek et al. recently reported in a large multicenter cohort of patients at high risk of PDAC that half of the patients who developed high grade dysplasia or cancer had no prior lesions detected by imaging 12 months before diagnosis [85].

Figure 3 shows examples of CT-scan aspects of pseudo-tumorous pancreatitis and para-duodenal pancreatitis.

Table 3 shows the different radiological signs that could be identified as discriminating between pseudotumorous CP (also called mass-forming pancreatitis) and PDAC but also between certain differential diagnoses (autoimmune pancreatitis, paraduodenal pancreatitis, obstructive CP) and PDAC. These are signs concerning the parenchyma, pancreatic ducts, and adjacent anatomical elements. Some signs have been evaluated for their diagnostic value, in particular the “duct-penetrating sign” which is a reliable sign of benign diseases (specificity 96%; sensitivity 85%) and the “double duct sign” theoretically absent in benign disease and present in peri-ampullary PDAC (specificity 63–80%; sensitivity 50–76%) [79].

Figure 4 shows examples of CP-MRI aspects of CP, obstructive CP, and PDAC.

#### 4.3.2. Endoscopic Ultrasound

EUS-guided fine-needle aspiration biopsy (FNAB) is the technique that should provide the best information but it is hampered by the very hypo-echogenic nature of the parenchyma and the amount of calcification [86]. In fact, cancer develops in advanced forms (more than 10 years old) which are often very calcified. AI systems using deep learning analysis have been proposed to enhance the detection of PDAC during EUS [87]. The EUS must be undertaken outside of any inflammatory outbreak and is aimed at detecting suspicious-looking tissue areas or lymphadenopathies that will be punctured. Attention should also be paid to the appearance of biliary or pancreatic stenosis. Transgastric fine-needle aspiration biopsy of an infiltrative tissue area close to the celiac trunk may also be helpful if neoplastic cells are detected. Certain areas of tissue free of calcifications (and surrounded by calcifications) may appear more suspicious, especially if the parenchyma is not very indurated at puncture compared with the pancreas in the case of CP, which is often difficult to remove.

FNAB can solve the diagnostic problems, but the results are less good than for classical adenocarcinoma because of the difficulties in targeting lesions among cysts and calcifications and the already hypoechoic and heterogeneous appearance of the CP pancreas on EUS (Figure 3). The performance of EUS-FNAB in differentiating CP from cancer developed on CP tissue was evaluated prospectively in a single-center or retrospective multi-center setting. Sensitivity ranges from 75 to 85% with a good negative predictive value of 85 to 95% [88,89,90]. Other techniques coupled with EUS have been tested, such as contrast ultrasound or elastometry, but these are single-center studies for procedures that are otherwise operator dependent. Contrast ultrasonography seems to provide evidence of the more vascular nature of pseudotumoral CP compared with cancer, but there is little data on elastometry [91,92,93,94].

Among the complementary techniques of the EUS-FNAB, molecular analysis can help the diagnosis in case of doubt on the cytopathological analysis of the biopsy material. The major somatic gene mutation in pancreatic cancer is the *KRAS* oncogene. *KRAS* mutation testing is clinically applicable. Indeed, this research is possible by gene amplification in pancreatic biopsies. We have been developing this test for 20 years, which is routinely performed on the rinsing of the puncture needle after recovery of the biopsy core. The material is stored in a special medium (RNAprotect Cell) until DNA and RNA are extracted. We can obtain 30 ng or more of good quality DNA for *KRAS* mutation analysis by Taqmann and more recently by next generation sequencing (NGS). In addition, NGS allows us to search for mutations in other genes, such as *BRCA1, BRCA2,* and *NTRK*, which is useful in the context of targeted therapies.

This search for *KRAS* mutations in cytological material has been validated by numerous prospective studies, including notes in two indications: the differential diagnosis between pseudotumor CP and cancer developed on CP and in the diagnosis of cancer. Indeed, the presence of the mutation is highly suggestive of cancer in cases where the sample is non-contributory or doubtful.

In contrast to pancreatic juice, *KRAS* mutations are highly infrequent in pancreatic tissue of CP. In the absence of a mutation, the diagnosis of cancer can therefore be ruled out in the case of pseudo-tumorous CP. Furthermore, NGS allows the simultaneous detection of mutations in *TP53*, *INK4a*, and *DPC4*, thus enriching the molecular diagnosis. Conversely, the presence of the *KRAS* oncogene mutation will argue for carcinomatous transformation on CP [11,90,95,96,97,98]. An example is given in Figure 5.

As for the cancer itself, in the presence of a suggestive clinical and radiological picture, even if the biopsy is non-contributory or doubtful, the presence of the *KRAS* mutation (and other target genes) will be highly suggestive of adenocarcinoma (the diagnostic values of the biopsy have been greatly improved, especially at the expense of the negative predictive value). This will avoid further investigations and biopsies and move the patient more quickly into the management process [11,90,95,96]. This point is illustrated in Figure 4 showing a case of pseudo-tumorous CP with suspected PDAC developed on a known CP and an FNAB non-contributive but positive for *KRAS* and *TP53* mutation: an adenocarcinoma was finally found on the resected pancreatic specimen.

### 4.4. Role of Liquid Biopsy

Pilot studies have been conducted in PDAC patients to assess the diagnostic value of detecting circulating tumor cells, circulating tumor DNA (cDNA), exosomes, and tumor-derived platelets. The main target for identification of cDNA and transport DNA in exosomes is the demonstration of the *KRAS* mutation [11]. Although digital droplet PCR (ddPCR) and NGS techniques are very sensitive, the detection of the circulating *KRAS* mutation imperfectly reflects the mutational load of the primary tumor from which it originates. However, it reflects the stage of the tumor quite well. The presence of a *KRAS* mutation in the cDNA is observed in 70–80% of patients with locally advanced and metastatic disease, whereas this value is often lower for patients with resectable tumors. Nevertheless, sensitivity appears to be better with ddPCR (43 to 78%) compared to conventional PCR or sequencing (27 to 47%) [11,99,100,101,102,103].

Most of the studies have included mainly pancreatic cancer and the control groups (when they were representative enough) have not necessarily included CP patients. Moreover, if small PDAC tumors do not release sufficient tumor cells and/or mutated DNA for *KRAS*, it is difficult to extrapolate and apply this method of diagnosis in the specific case of PDAC developed on CP tissue. One possible approach might be the combination of several methods for detecting circulating tumor elements, including searching for circulating tumor cells, exosomes, and tumor-educating platelets [104,105,106]

MicroRNAs (miR) are also promising markers and we have already demonstrated the critical role of miR-21, miR-148a, Let7b, and miR-200 in pancreatic carcinogenesis and the promising role of miR assays in human fluids for the positive and differential diagnosis of PDAC [107,108,109,110,111,112,113]. Recent studies have explored the diagnostic value of circulating miR in the particular clinical context of diagnosis of PDAC versus CP or human healthy controls. The first pilot study from Vila-Navarro et al. found that the plasma level of nine miR was significantly increased in CP patients when compared to control plasmas [114]. A second study showed significant upregulation of miR-215-5p, miR-122-5p, and miR-192-5p in PDAC serum samples. In contrast, levels of miR-30b-5p and miR-320b were significantly lower in PDAC compared to CP and controls [115]. Another study has revealed an overexpression of miR-200b and miR-200c in serum exosomes of PDAC patients compared to healthy controls and patients with CP [116]. Finally, Wang L et al. observed that plasma-derived exosomes-miR-19b levels in PDAC patients were significantly higher than those in patients with non-PDAC pancreatic tumors, CP patients, and healthy volunteers [117]. On the whole, if a specific signature of PDAC and PC can be validated in future large-scale studies (within exosomes and/or plasma extracellular vesicles), these will be tools applicable to the particularly difficult context of the diagnosis of PDAC developed on CP.

### 4.5. Role of Surgery

Imaging should be used unsparingly, with CT, CP-MRI, and EUS sequences with FNAB (as well as molecular biology) to distinguish between inflammation and possible adenocarcinoma (algorithm presented in Figure 6). In all these difficult cases, it will be essential to discuss patients’ records during a multidisciplinary consultation meeting or one specializing in pancreatology, if possible, in order to establish the diagnostic and therapeutic features and above all the action to be taken in the event that no decision is taken or if the diagnosis is uncertain. Indeed, sometimes evolution and the passage of time help to resolve the issue, especially if the records are examined in proximity to inflammatory outbreaks whose lesions will blur the lines between inflammation and a genuine tumor. The surveillance period and methods are important in order not to miss a curable and/or operable cause. At the end of this entire sequence of investigations, surgical exploration and hopefully resection will be part of the final approach where the slightest doubt exists in order to potentially treat a small adenocarcinoma at a curable stage.

## 5. Can Pancreatic Cancer Developed in Chronic Pancreatitis Be Detected and Prevented?

As far as alcoholic CP is concerned, the low frequency of occurrence of adenocarcinoma does not dictate a systematic screening approach whatever the stage of the disease. However, once again, if there is the slightest doubt, close clinical and radiological (or even EUS) surveillance is required to confirm or exclude cancer in CP.

In the specific context of hereditary pancreatitis due to the *PRSS1* gene mutation, it is recommended that cancer screening be carried out from the age of 40 [118]. Individuals with an autosomal dominant history of hereditary pancreatitis but without identified mutation of PRSS1 are also to be included in this annual screening. This surveillance should be performed’ in expert centers with pancreatic specialists. The international consensus guidelines on surveillance of pancreatic cancer in chronic pancreatitis do not recommend screening for patients with *SPINK1* p. N34S or with other germline mutations including those of *CFTR, CTRC, CPA1,* and *CEL* based on the rationale that PDAC risk in this subpopulation is not high enough to support annual surveillance [118]. Screening should be discussed in the case of morphological signs of CP and depending on whether associated adenocarcinoma risk factors exist (diabetes, smoking, obesity, family history of pancreatic adenocarcinoma in a first degree relative).

CP-MRI is the reference examination because it is non-irradiating but must be completed by a CT scan if the MRI is technically deficient (patient movements, very atrophic pancreas of difficult analysis) or EUS in case of absence of calcifications. The latter is indeed sensitive for the detection of small lesions when applying high frequencies (10 to 12 MHz). It can also be considered to alternate with MRI. In the event of an abnormality, monitoring can be repeated every three to six months (see National Diagnostic and Care Protocol at https://www.has-sante.fr/jcms/p_3225352/fr/pancreatite-chronique-hereditaire) (accessed on 15 November 2020). However, the determination of tumor markers carcino-embryonic antigen (CEA) and carbohydrate antigen 19-9 (CA 19-9) is not recommended at this follow-up.

As far as prevention is concerned, it is therefore essential to obtain early and definitive smoking cessation in these patients with, needless to say, abstinence from alcohol or the use of other toxic substances such as cannabis [119]. The international consensus guidelines also recommend a healthy diet containing daily fruit and vegetables with a high folate intake, whilst moderating the intake of red meat and taking some form of regular intense physical exercise, all with a view to preventing obesity [118].

## 6. Conclusions

Although rare, the development of cancer in chronic pancreatitis must be kept in mind by the clinician, particularly in the context of CP where clinical signs evolve and become concerning, such as back pain, weight loss, unbalanced diabetes, and jaundice. Imaging is difficult to interpret and must be carefully reviewed, bearing in mind that if there is the slightest doubt, EUS-FNAB may be of assistance by targeting suspicious areas within inflammatory, fibrotic, and calcified parenchyma with the help of molecular biology (search for *KRAS* oncogene mutation). Short-term follow-up of patients is necessary at the clinical and paraclinical level to try to diagnose cancer at a surgically curable stage or to establish differential diagnoses, such as authentic pseudotumor chronic pancreatitis, and whether or not this is associated with para-duodenal pancreatitis.

## Figures and Tables

**Figure 1 cancers-15-00761-f001:**
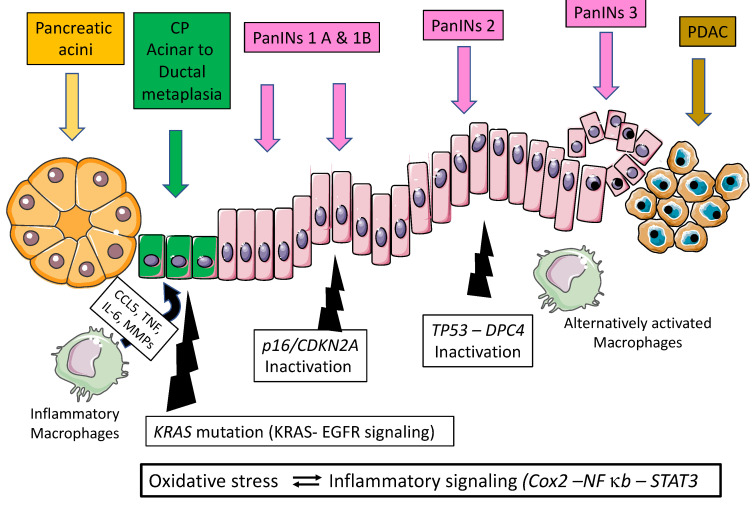
Natural history of pancreatic carcinogenesis in the specific context of inflammation.

**Figure 2 cancers-15-00761-f002:**
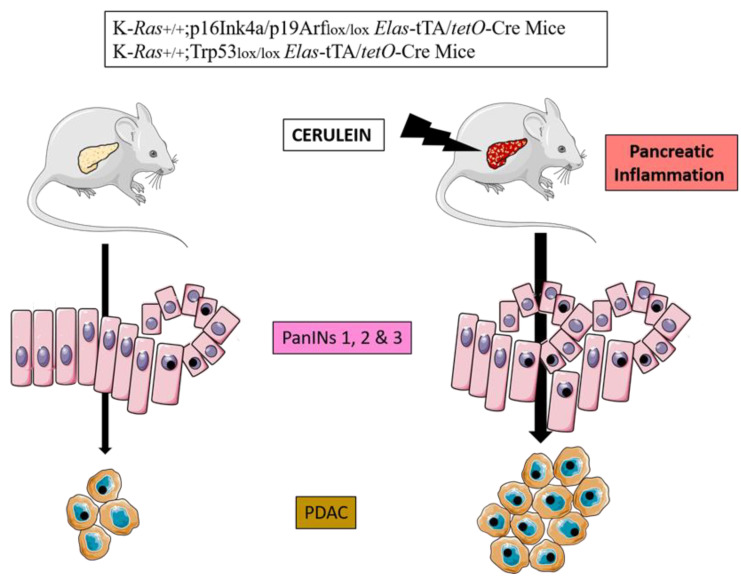
Cerulein experimental model of pancreatic carcinogenesis in genetically engineered mice.

**Figure 3 cancers-15-00761-f003:**
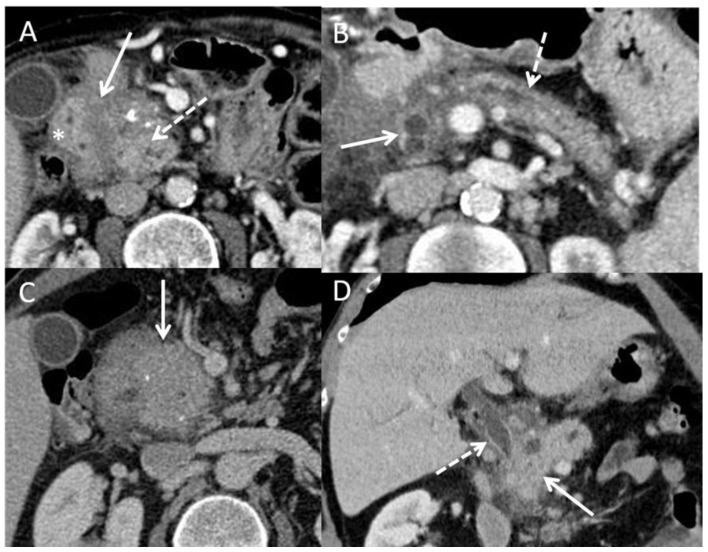
CT scan pictures of paraduodenal pancreatitis and pseudo-tumorous chronic pancreatitis. (**A**): paraduodenal pancreatitis developed on chronic pancreatitis. Hypodensity of the duodenal groove (white arrow) and hypertrophy of the calcified head of pancreas (dashed white arrow) (white star: duodenal wall). (**B**): paraduodenal pancreatitis with duodenal cysts (white arrow) associated with chronic pancreatitis with irregular caliper of Wirsung duct (white dashed arrow); (**C**): pseudotumorous chronic pancreatitis on the head of the pancreas (white arrow) without vessel infiltration; (**D**): pseudotumorous chronic pancreatitis on the head of the pancreas (white arrow) with stenosis of the common bile duct (white dashed arrow).

**Figure 4 cancers-15-00761-f004:**
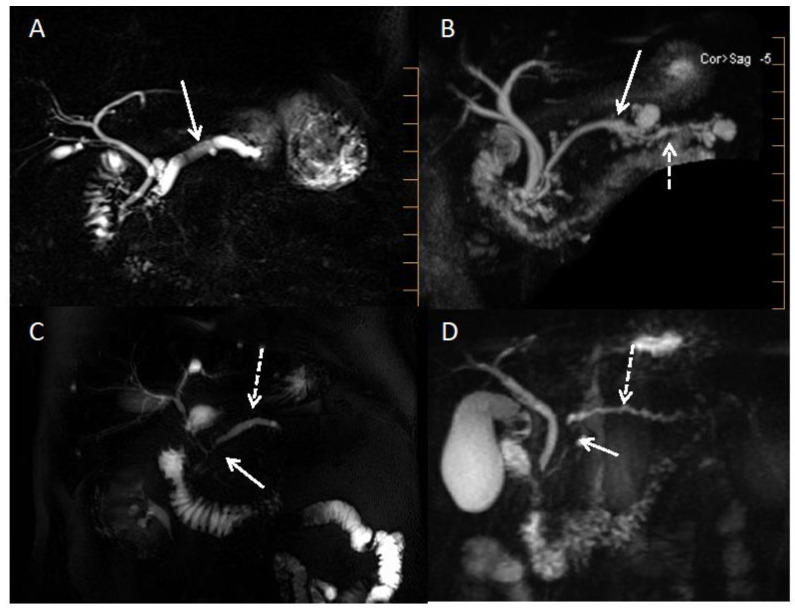
MRI cholangiopancreatography pictures of alcoholic chronic pancreatitis, pancreatic carcinoma, and obstructive chronic pancreatitis. (**A**): Alcoholic chronic pancreatitis with dilatated Wirsung duct and one or two pancreatic lithiasis (white arrow). (**B**): Mixed IPMN (white arrow) with upstream signs of obstructive chronic pancreatitis (dashed white arrow). (**C**): Pancreatic ductal adenocarcinoma (tumoral stenosis of the isthmus showed by the white arrow) with upstream regular Wirsung duct dilatation (dashed white arrow). (**D**): Pancreatic ductal adenocarcinoma (tumoral stenosis of the head showed by the white arrow) with upstream signs of obstructive chronic pancreatitis (dashed white arrow).

**Figure 5 cancers-15-00761-f005:**
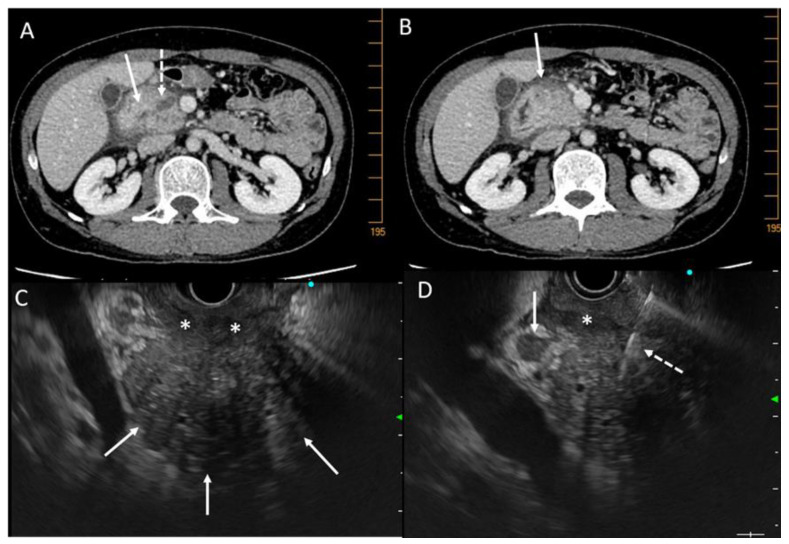
CT scan and EUS pictures of a case of pancreatic carcinoma developed on chronic pancreatitis. A 47-year-old man with alcoholic chronic pancreatitis and recurrence of abdominal pain despite alcohol abstinence. (**A**): CT scan pictures of tissue mass on the head of the pancreas (white arrow) and dilatation of Wirsung duct (dashed white arrow); (**B**): tissue infiltration at the anterior part of the head of the pancreas (white arrow); (**C**): EUS view of the hypertrophic head of the pancreas (white arrows) with tissue infiltration of the duodenum (white stars); (**D**): EUS view of the hypertrophic head of the pancreas with lymphadenopathy (white arrow) and fine-needle aspiration biopsy (white dashed arrow) which was non-contributive but displayed *KRAS* mutation. A carcinoma was found on the surgical resected specimen.

**Figure 6 cancers-15-00761-f006:**
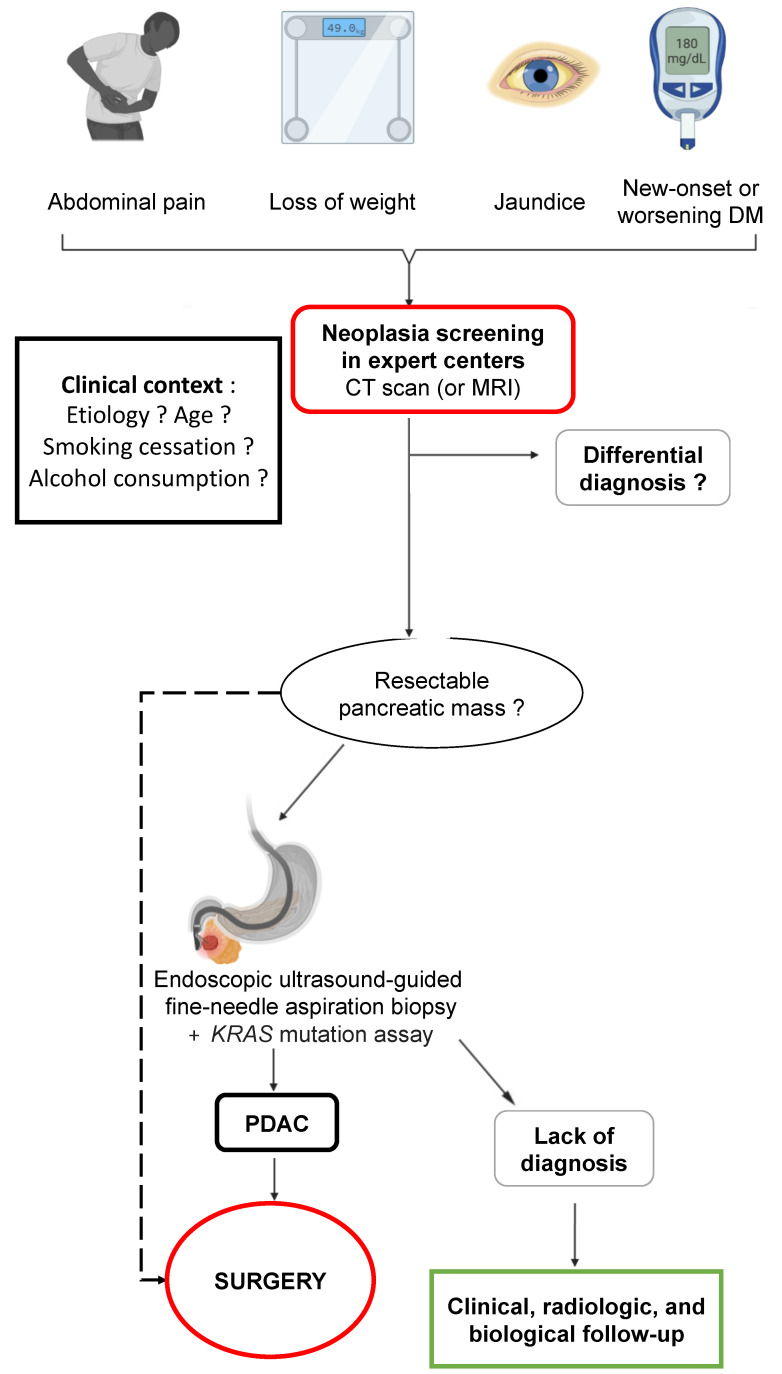
Proposed screening algorithm for pancreatic ductal adenocarcinoma in patients with chronic pancreatitis. DM: diabetes mellitus; CT scan: computed tomography scan; MRI: magnetic resonance imaging; PDAC: pancreatic ductal adenocarcinoma.

**Table 1 cancers-15-00761-t001:** Main genetically engineered mouse models demonstrating the role of inflammation in pancreatic carcinogenesis.

Authors, Year, [Ref.]	Genetically Engineered Model	Pathway Affected and Phenotype
Muller-Decker et al., 2006 [24]	*Krt5-Cox2^Tg^*	Cox2 overexpression: chronic pancreatitis and ductal neoplastic lesions
Al Saati et al., 2013 [29]	*Pdx1-Cre;Kras^G12D^;TP53INP1-/-*	Oncogenic *RAS* + oxidative status dysregulation: accelerated PanINs formation
Daniluk et al., 2012 [30]	*Ela-Cre^ERT^;Kras^G12D^;Cox^Tg^*	Oncogenic *RAS* + Cox2 overexpression: rapid development of chronic inflammation and PanINs
Maniati et al., 2011 [26]	*Pdx1-Cre;Kras^G12D^;IKK2 ^FI/FI^*	Oncogenic *RAS* + NfκB pathway inhibition: impaired PanIN formation and decreased PDAC development
Daniluk et al., 2012 [30]	*Ela-Cre^ERT^;Kras^G12D^;IKK2^Tg^*	Oncogenic *RAS* + NfκB pathway activation: increased fibrosis and rapid development of PanINs
Lesina et al., 2011 [31]	*Ptf1a-Cre^Ex1^;Kras^G12D^;Socs3 ^FI/FI^*	Oncogenic *RAS* + Stat3 activation: accelerated PanIN progression and increased PDAC formation
Guerra et al., 2011 [32]	*Kras^G12V^;p16Ink4a/p19arf ^Iox/lox^;Ela-tTA/tetO-Cre* *Kras^G12V^;Trp53 ^Iox/lox^;Elas-tTA/tetO-Cre*	Oncogenic *RAS* + loss of *p16Ink4a/p19arf* or *Trp53* + cerulein injections: increased PDAC formation and progression
Liou et al., 2016 [33]	*Ptf1a/p48^Cre/+^* *LSL-Kras^G12D/+^* *PKD1^fl/fl^*	Treatment of mice by mitochondria-targeted antioxidant MitoQ: reduced *KRAS*-induced formation of ROS with reduced formation of PanINs

Cre: cre-Lox recombination; Ela: Elastase; tetO: tetracycline operator; TRE: tetracycline response element; LSL: lox-Stop-lox; IKK2: nuclear NfκB inhibitor; Ptf1a: pancreas associated transcription factor 1a; PKD1: protein kinase D1.

**Table 2 cancers-15-00761-t002:** Pancreatic cancer risk according to the etiology of chronic pancreatitis.

Etiology	Pancreatic Cancer Estimated Risk
Alcoholic CP	Incidence of 2 and 4% after 5 and 20 years of evolution, respectively.
Hereditary pancreatitis (*PRSS1*)	Incidence of 10, 19, and 53.5% at 50, 60, and 75 years, respectively.
*SPINK1* mutations	Incidence of 12, 28, and 52% at 60, 70, and 80 years.
*CFTR* mutations	Increased risk of PDAC of 1.41 compared to control patients.
*CTRC, CASR, CLDN2, CPA1, TRPV6, CEL-HYB* mutations	No available data due to very low incidence of these mutations.

CP: chronic pancreatitis; PDAC: pancreatic ductal adenocarcinoma.

**Table 3 cancers-15-00761-t003:** Predictive value of radiological signs.

	Signs Evocative of Benign Disease	Signs Evocative of Malignant Disease
Parenchymal signs	*Pseudotumorous CP and IPMN:* - Absence of enhancement of the mass - Hypointense T1; iso to hyperintense T2 - Moderate enhancement at portal phase after gadolinium	- Late enhancement of the mass at portal phase (CT and gadolinium MRI) - Parenchyma atrophy associated with Wirsung dilation - Displaced parenchymal calcifications (mass pushing calcifications at periphery of the gland)
	*Paraduodenal pancreatitis:* - Parietal thickening of the second part of the duodenum on the pancreatic side - One or more cystic images inside the sulcus and in the duodenal wall - Absence of gland atrophy	
	*AIP:* - Loss of lobulation (“sausage”-shaped pancreas) - Peripancreatic hypodense rim	
Duct signs	*Obstructive CP and IPMN:* - Duct-penetrating sign (mass penetrated by an unobstructed pancreatic duct) - Branch ducts dilation	- Complete obstruction of the Wirsung duct at the mass - Double duct sign (dilation of Wirsung and bile duct)
Vessels signs	*Pseudotumorous CP:* - Displaced vessels - Fat separating the mass and vessels	- Infiltration of the vessels (particularly arterial) and the celiac region
Other signs	*AIP:* - Other organs involved (kidneys, aorta, etc.)	- Peripancreatic tissue infiltration

CP: chronic pancreatitis; PDAC: pancreatic ductal adenocarcinoma; AIP: auto immune pancreatitis; IPMN: intraductal papillary mucinous neoplasms.

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
