# Peer review of "Pancreatic Cancer in Chronic Pancreatitis: Pathogenesis and Diagnostic Approach"

_cancers, 2023, doi:10.3390/cancers15030761_

Round 1

Reviewer 1 Report

The title of the work is not adequate to the content of the work. The review in its form is not coherent.

In the initial part of the paper, the authors focus on the genetic background of pancreatic cancer carcinogenesis, mainly on the K-ras pathway, including studies on mouse models of PDAC etc. What they fail to mention, is the exceptional complexity of the PDAC. For many years, numerous studies have been conducted focusing on the search for the specific and sensitive diagnostic and prognostic marker of pancreatic cancer, but their results are still unsatisfactory. In PDAC there is genetic instability consisting in the accumulation of numerous mutations and chromosomal aberrations. It is estimated that in advanced cancer there are an average of 63 mutations grouped in about 12 signaling pathways, including i.a. Notch, TGF-β, K-ras and Hedgehog.  In the course of long-term CP, irreversible morphological disorders of the pancreas occur, as well as gradual fibrosis of the parenchyma, therefore early focal neoplastic changes may not be visible in the pancreatic parenchyma changed in this way. On the other hand, recurrent CP exacerbations may lead to the formation of benign inflammatory tumors, which may raise the suspicion of a malignant lesion. Early diagnosis of PDAC in patients with long-term CP is particularly difficult. So far, there are no non-invasive methods to distinguish them.

The stages of carcinogenesis on the basis of inflammation are well described and presented.

In my opinion, it is not necessary to describe the significance of genetically modified mouse models, unless the topic of the paper is changed. In the first part the review is too detailed (especially with basic sciences) and in the second part it is too fuzzy .

Chapter no. 4.1. „Clinical signs”  requires significant corrections f.ex.: after ":" there are no bullet points listed, only capitalized sentences. Morover, the symptoms of PDAC (mostly advanced) have been only briefly described and mixed with situations that strongly predispose to accelarated development of PDAC.

Then, in the next chapter „4.2.1. Standard Biology”, the authors describe in detail the results of individual studies on markers that increase the sensitivity of CA19-9.

Then there are several subchapters discussing imaging diagnostics without specific values ​​of sensitivity, specificity, their improvement, including types of EUS needles, number of punctures, etc., without providing guidelines for performing specific tests in reference centers according to the pancreatic protocol.

Then, there are again too detailed chapters 4.2.5. Endoscopic ultrasound-guided fine needle biopsy coupled with molecular biology and 4.2.6. Role of liquid biopsy. The authors focus, one again,  on te K-ras path , co maybe the tittle of the work should be completely changed and another chapters, poorly described, should be deleted from the review.

The chapter 4.3. „Particular forms of difficult diagnosis” adds nothing to the work and repeats informations that are described in previous ones.

The authors did not attempt to consider how to recognize the PDAC earlier so that it can be operated. The symptoms in the algorithm shown in Figure 6 are symptoms of advanced PDAC. Already at the stage of diagnostics, the patient should be referred to a reference center for CT scan to be performed according to the pancreatic protocol by an experienced radiologist.

In chapter 4.4. Summary of the diagnostic approach and role of surgery , authors once again repeat the same informations.

The review should be much shortened and organized.  Authors should think about what topic they care about and be more consistent in form.

Author Response

We thank the reviewer for his/her constructive remarks. We have revised the manuscript by reorganizing the clinical part which has been clarified, simplified and reduced to balance the text. We have left the "carcinogenesis and animal models" part after modifying the title of the Ms which finally includes two parts on the subject "pathogenesis and diagnostic approach". The first part is original because the synthesis of those models that specifically link inflammation and pancreatic cancer is rarely presented (we have notably added an original model suggested by the reviewer 2).

The main revisions made are:

- Changing the title, which is now more explicit

- Addition of a sentence at the beginning of chapter 2.1 presenting the hallmark of mutations observed during PDAC (plus new reference 12)

- Chapter 4 has been better organized by distinguishing now "4.1 clinical signs", 4.2 biology, 4.3 imaging, 4.4 liquid biopsy, 4.5 surgery”

- Subchapter 4.3 is simply divided into 2 parts "ultrasonography, CT-scan and MRI" and "endoscopic ultrasound”

- To avoid redundancy, the subchapter EUS-FNAB and molecular biology has been integrated now into the EUS chapter.

- Similarly, the 3 former chapters on differential diagnosis (former chapters 4.3.1, 4.3.2 and 4.3.3) have been deleted and inserted in a very reduced form in the subchapter "imaging".

- In addition, we have added a new table (table 3) that details the radiological signs that can help to differentiate between CP and PDAC but also all the differential diagnosis (AIP, paraduodenal pancreatitis, obstructive CP) and PDAC (plus new references 77 to 83)

- Values of sensitivity and specificity have been added also. In the text.

- The figures 3, 4 and 5 have been repositioned according to the new organization of this chapter

- By these revisions we fully agree that the manuscript after reorganization and reduction in size is clearer and directly exposes the issues of PDAC diagnosis developed from PDAC

- Figure 6 has been modified by specifying in a box that the imaging must be done in an expert center.

Reviewer 2 Report

Dear Authors,

The selected topic was quite interesting and there a lot of scope to work on chronic pancreatitis for the therapy.

The authors has covered about the Genetics of pancreatic carcinogenesis via KRAS mutations and the pathway related to KRAS.

The authors can also include about the NLRP3 inflammasome and its role related to Caspase-1 and IL-18 pathway.

Cerulein  induced pancreatic carcinogenesis in genetically engineered mice was well represented.

Paraclinical tests and imaging using Ultrasonography, Tomodensitometry (CT scan) and MRI to detect the PDAC was clearly presented in the article.

The authors can add few more references 

1. Kandikattu, H. K., Venkateshaiah, S. U., Yadavalli, C. S., Oruganti, L., & Mishra, A. (2022). Development and Characterization of Novel Chronic Eosinophilic Inflammation-Mediated Murine Model of Malignant Pancreatitis. Endocrine, Metabolic & Immune Disorders Drug Targets.

2. Kandikattu, H. K., Manohar, M., Venkateshaiah, S. U., Yadavalli, C., & Mishra, A. (2021). Chronic inflammation promotes epithelial-mesenchymal transition-mediated malignant phenotypes and lung injury in experimentally-induced pancreatitis. Life Sciences278, 119640.

Author Response

We thank the reviewer for his/her constructive comments.

We have added the role of NLRP3 inflammasome and associated signaling in chapter 2.2 (Plus two new references 21 and 22)

The model developed in mice by injection of cerulein plus azoxymethane has been added at the end of the chapter 2.3 (plus two new references 27 and 28).

Round 2

Reviewer 1 Report

This paper is a very good, insightful summary of the pathogenesis of PDAC on the basis of chronic inflammation and  diagnostic imaging techniques in context of differentiation from benign pancreatic lesions. Changing the topic and introducing additional data, e.g. table no. 3, made the work very coherent.